# Recruitment of Muscle Genes as an Effect of Brown Adipose Tissue Ablation in Cold-Acclimated Brandt’s Voles (*Lasiopodomys brandtii*)

**DOI:** 10.3390/ijms24010342

**Published:** 2022-12-25

**Authors:** Min Liu, Xue-Ying Zhang, Chen-Zhu Wang, De-Hua Wang

**Affiliations:** 1State Key Laboratory of Integrated Management of Pest Insects and Rodents, Institute of Zoology, Chinese Academy of Sciences, Beijing 100101, China; 2CAS Center for Excellence in Biotic Interactions, University of Chinese Academy of Sciences, Beijing 100049, China; 3School of Life Sciences, Shandong University, Qingdao 266237, China

**Keywords:** skeletal muscle, brown adipose tissue (BAT), interscapular BAT (iBAT) removal, cold acclimation, nonshivering thermogenesis (NST)

## Abstract

Skeletal muscle-based nonshivering thermogenesis (NST) plays an important role in the regulation and maintenance of body temperature in birds and large mammals, which do not contain brown adipose tissue (BAT). However, the relative contribution of muscle-based NST to thermoregulation is not clearly elucidated in wild small mammals, which have evolved an obligate thermogenic organ of BAT. In this study, we investigated whether muscle would become an important site of NST when BAT function is conditionally minimized in Brandt’s voles (*Lasiopodomys brandtii*). We surgically removed interscapular BAT (iBAT, which constitutes 52%~56% of total BAT) and exposed the voles to prolonged cold (4 °C) for 28 days. The iBAT-ablated voles were able to maintain the same levels of NST and body temperature (~37.9 °C) during the entire period of cold acclimation as sham voles. The expression of uncoupling protein 1 (UCP1) and its transcriptional regulators at both protein and mRNA levels in the iBAT of cold-acclimated voles was higher than that in the warm group. However, no difference was observed in the protein or mRNA levels of these thermogenesis-related markers except for PGC-1α in other sites of BAT (including infrascapular region, neck, and axilla) between warm and cold groups either in sham or iBAT-ablated voles. The iBAT-ablated voles showed higher UCP1 expression in white adipose tissue (WAT) than sham voles during cold acclimation. The expression of *sarcolipin* (*SLN*) and *sarcoplasmic endoplasmic reticulum Ca^2+^-dependent adenosine triphosphatase* (*SERCA*) in skeletal muscles was higher in cold than in warm, but no alteration in *phospholamban* (*PLB*) and phosphorylated-PLB (P-PLB) was observed. Additionally, there was increased in iBAT-ablated voles compared to that in the sham group in cold. Moreover, these iBAT-ablated voles underwent extensive remodeling of mitochondria and genes of key components related with mitochondrial metabolism. These data collectively indicate that recruitment of skeletal muscle-based thermogenesis may compensate for BAT impairment and suggest a functional interaction between the two forms of thermogenic processes of iBAT and skeletal muscle in wild small mammals for coping cold stress.

## 1. Introduction

Thermogenesis is an important homeostatic mechanism essential for survival and normal physiological functions in endotherms including mammals and birds, which guarantees evolutionary expansion [1]. Muscle shivering is an immediate response to cold stress, however, continuous muscle shivering leads to exhaustion and muscle damage [1]. Therefore, nonshivering thermogenesis (NST) mechanisms are activated in these conditions for long-term thermal homeostasis. Studies in birds, which do not contain brown adipose tissue (BAT), have demonstrated the importance of skeletal muscle-based NST [2]. But the controversial existence of NST in birds cannot be used to demonstrate the existence of muscle NST in mammals. Most of small mammals, however, have evolved an obligate thermogenic organ of BAT [3,4,5].

Brown adipocytes express uncoupling protein 1 (UCP1) in the inner membrane of mitochondria, which, upon activation, depletes the proton gradient (or uncouples oxidative phosphorylation) and generates heat rather than ATP [6,7,8]. Beige adipocytes are an induced form of thermogenic adipocytes that appear in white adipose tissue (WAT) after various external stimuli, such as cold exposure [9]. A study has shown that *UCP1* knockout (*UCP1*^−/−^) mice were sensitive to acute cold exposure but could be gradually acclimated to long-term cold (4 °C) [10], which suggests the existence of other sites for NST. It has also been speculated that, in addition to the proton leakage promoted by UCP1, BAT sarcoplasmic endoplasmic reticulum (SER) Ca^2+^-dependent adenosine triphosphatase (SERCA) may represent one of the thermogenic pathways contributing to the thermogenic function of brown adipocytes [11]. However, it has been demonstrated that ATP-dependent Ca^2+^ cycling enhanced by SERCA2b and type 2 ryanodine receptor (RyR2) in beige adipocytes plays a powerful thermogenic mechanism in the absence of UCP1 [12]. Several studies in mice have suggested that in addition to BAT, skeletal muscle has a role in NST [13,14,15], but the molecular details of its involvement have not been completely explored.

Studies have shown that uncoupling of SERCA by sarcolipin (SLN) plays an important role in muscle NST and that loss of SLN impairs core body temperature (*T*_b_) maintenance in mice [16,17,18]. The activity of SERCA in muscle is regulated by two different regulatory proteins, phospholamban (PLB) and SLN [19]. But researchers challenged the mice with acute cold (4 °C) and found that *PLB*^−/−^ mice did not develop hypothermia [17]. The type 1 ryanodine receptor (RYR1) is a Ca^2+^-release channel in the SER of skeletal muscle [20], the study in mice showed that RYR1-mediated Ca^2+^ leak is an important mechanism for SERCA-activated heat generation [16]. At present, most studies on skeletal muscle thermogenesis in rodents focus on mice, but few studies concentrate on wild rodents.

Brandt’s voles (*Lasiopodomys brandtii*) are small non-hibernating herbivorous rodents that are widely distributed in the dry steppe zone of Mongolia, the southeast of Baikal region of Russia and the Inner Mongolian grasslands of China [21], and the temperature of its habitat in winter can reach −25–−40 °C [22]. We previously found that resting metabolic rates (RMR) was 15% higher than that of the control, and UCP1 content in iBAT showed a 43% increase at the end of 4-week cold acclimation in Brandt’s voles [23]. Previous studies on NST in small mammals mainly focused on the interscapular BAT (iBAT), however, no attention has been paid to whether the BAT located in other sites including infrascapular region, neck, and axilla has the same function as iBAT, and whether the skeletal muscle contributes to cold-induced thermogenesis in wild small mammals. The main purpose of this study was to determine whether the skeletal muscles of Brandt’s voles produce heat at low temperature and whether they can compensate for thermogenesis when BAT function is minimized. We hypothesized that skeletal muscles contributed to cold-induced thermogenesis and could compensate for impaired BAT thermogenic function in wild rodents.

## 2. Results

### 2.1. IBAT Ablation Did Not Affect Basal or Adaptive Thermogenic Capacity

Throughout the period of temperature acclimation in the voles, there was no significant difference in body mass among groups (F_3,23_ = 1.654, *p* = 0.205, Figure 1A), but there was a significant difference in food intake in four groups (F_3,23_ = 11.560, *p* < 0.001, Figure 1B). Low temperature increased food intake significantly (F_1,23_ = 33.702, *p* < 0.001), but iBAT removal had no significant effect (F_3,23_ = 0.029, *p* = 867).

Neither low temperature (F_1,25_ = 0.627, *p* = 0.436) nor iBAT removal (F_1,25_ = 0.126, *p* = 0.726) affected rectal temperature in the last week of acclimation (Figure 1C). Cold acclimation resulted in the increase of RMR (F_1,25_ = 12.230, *p* = 0.002) and NST (F_1,25_ = 74.826, *p* < 0.001). In the removal groups, cold induced RMR 27.2% higher and NST 58.0% higher than warm, but in the sham groups, cold induced RMR 22.0% higher and NST 45.6% higher than warm_._ IBAT removal had no significant effect on RMR (F_1,25_ = 0.407, *p* = 0.529) or NST (F_1,25_ = 1.330, *p* = 0.260).

### 2.2. IBAT Ablation Did Not Induce UCP1-Dependent Thermogenesis in other Sites of BAT

A small amount of new BAT regrew only in some individuals (3/6 in cold removal (CR) group, ~9.4% of total iBAT; and 4/6 in warm removal (WR) group) and no difference was observed in the two removal groups (t = −0.290, df = 5, *p* = 0.783. Figure 2A). In order to confirm the role of BAT in the thermogenesis of Brandt’s voles, we detected the mRNA expression and protein levels both in iBAT and other parts of BAT (infrascapular region, neck and axilla). For iBAT, cold acclimation increased mRNA expression of *UCP1* (*p* = 0.022), *peroxisome proliferator-activated receptor gamma coactivator 1-alpha* (*PCG-1α*, a cotranscription factor for mitochondriogenesis, *p* = 0.046), and *proliferating cell nuclear antigen* (*PCNA,* a marker of cell proliferation, *p* = 0.003), but had no effect on *peroxisome proliferator-activated receptor gamma* (*PPARγ*, *p* = 0.454, Figure 2B), an indispensable transcription factor in the early differentiation of BAT. The protein levels showed similar patterns as mRNA expression during cold acclimation. Cold acclimation increased expression of UCP1 (t = 3.069, df = 10, *p* = 0.012), PGC-1α (t = 2.253, df = 10, *p* = 0.048), deiodinase iodothyronine type II (DIO2, thyroid hormone synthase, t = 2.746, df = 10, *p* = 0.021), and tyrosine hydroxylase (TH, the rate-limiting enzyme in the process of catecholamine biosynthesis, t = 2.381, df = 10, *p* = 0.039; Figure 2C) in iBAT. Cold and iBAT removal did not affect the expression of *UCP1* (F_3,20_ = 0.156, *p* = 0.925. Figure 2D), *PGC-1α* (F_3,20_ = 1.448, *p* = 0.259. Figure 2E), *PPARγ* (F_3,20_ = 0.236, *p* = 0.870. Figure 2F) or *PCNA* (F_3,20_ = 2.987, *p* = 0.056. Figure 2G), but *PCNA* expression in CR was significantly higher than in cold sham (CS, *p* = 0.026), WR (*p* = 0.019) and warm sham (WS, *p* = 0.029). However, both cold and iBAT removal had no significant effect on the protein levels of UCP1 (F_3,20_ = 1.241, *p* = 0.321. Figure 2H), PPARγ (F_3,20_ = 0.602, *p* = 0.621. Figure 2J) or PCNA (F_3,20_ = 0.932, *p* = 0.443, Figure 2K) in other sites of BAT, except for PGC-1α (F_3,20_ = 13.721, *p* < 0.001, Figure 2I). These results suggest that iBAT is the main part of BAT thermogenesis and iBAT removal does not induce a compensative increase of thermogenesis in other sites of BAT. At low temperature, total UCP1 in BAT of other parts was no significant difference in cold groups (t = 0.351, df = 10, *p* = 0.057), the result is displayed in the inserted panel of Figure 2H. 

### 2.3. Cold and iBAT Removal Induced Browning of WAT

The expression of UCP1 in WAT was detected to determine whether cold and iBAT removal would affect the browning of WAT. The immunohistochemical results of UCP1 reflected the browning degree of white fat. White adipocytes have larger fat droplets, while brown adipocytes have smaller (Figure 3A). Cold induced an increase of UCP1-positive ratio (F_1,20_ = 30.041, *p* = 0.001), but iBAT removal showed no significant effect (F_1,20_ = 3.908, *p* = 0.062). Under the condition of low temperature, the degree of browning in iBAT-ablated voles was significantly higher than that in sham group (LSD *post hoc* test, *p* = 0.033, Figure 3B), suggesting that iBAT ablation induced supplement in WAT browning.

### 2.4. Cold Acclimation and iBAT Removal Up-Regulated Expression of Major Ca^2+^-Cycling Proteins in Skeletal Muscle

To verify whether skeletal muscle-dependent thermogenesis increased during cold acclimation and after iBAT removal, we determined the expression of Ca^2+^-cycling mRNA or proteins in skeletal muscle. For mRNA expression, cold induced high *SERCA1* expression (F_1,20_ = 9.104, *p* = 0.007, Figure 4A). Cold (F_1,20_ = 3.590, *p* = 0.073) and iBAT removal (F_1,20_ = 3.366, *p* = 0.081) had no significant effect on the expression of *SERCA2* (Figure 4B), which in CR was significantly higher than in WS. Low temperature had a significant stimulative effect on *SLN* (a regulator of SERCA activity in muscle, F_1,20_ = 10.736, *p* = 0.004, Figure 4C), however the effect of iBAT removal was not significant. *PLB* expression was not affected by low temperature (F_1,20_ = 3.110, *p* = 0.093) or iBAT removal (F_1,20_ = 0.570, *p* = 0.459, Figure 4D). Neither temperature (F_1,20_ = 3.284, *p* = 0.085) nor removal (F_1,20_ = 1.824, *p* = 0.192) had a significant effect on *RYR1*, but there was an interactive effect between the two factors (*p* = 0.040), and the expression in CR was significantly higher than the other three groups (Figure 4E). The mRNA expression of *uncoupling protein 3* (*UCP3*) was not affected by temperature (F_1,20_ = 0.827, *p* = 0.374) or iBAT removal (F_1,20_ = 1.621, *p* = 0.218, Figure 4F). For protein levels, cold acclimation up-regulated SERCA1 (by 45.38%, F_1,20_ = 13.315, *p* = 0.002, Figure 4G), but IBAT removal showed no significant effect (F_1,20_ = 0.044, *p* = 0.836). Phosphorylated-PLB (P-PLB), a key regulator of SERCA, was not affected by the two factors (Figure 4H). Cold (F_1,20_ = 5.509, *p* = 0.029)increased expression of PGC-1α, but removal (F_1,20_ = 1.575, *p* = 0.224) had no significant effect. Cold (F_1,20_ = 0.827, *p* = 0.374) and iBAT removal (F_1,20_ = 1.621, *p* = 0.281) had no significant effect on the expression of UCP3 (Figure 4J), which is closely related to skeletal muscle thermogenesis. It suggested that cold acclimation and iBAT removal activated the regulation of SERCA for muscle thermogenesis.

### 2.5. Cold Acclimation and iBAT Removal Increased Mitochondrial Numbers and Metabolism-Related Genes in the Skeletal Muscle

We investigated whether iBAT ablation caused a substantial remodeling of mitochondrial architecture and metabolism-related genes in the skeletal muscles. Electron microscopic analyses of TA muscle (Figure 5A,B) revealed that both cold (F_1,12_ = 64.637, *p* < 0.001) and iBAT removal (F_1,12_ = 57.394, *p* < 0.001) induced a higher abundance of intramyofibrilar mitochondria. However, the images were not favorable, so cristae density could not be clearly seen. Mitochondrial metabolism was evaluated by measuring the expression of *citrate synthetase* (*CSY*), *acetyl-CoA carboxylase beta* (*ACACβ*) and *cytochrome C oxidase subunit II* (*COX II*). *CSY* expression at cold was significantly higher than at warm (F_1,20_ = 5.066, *p* = 0.036), but iBAT removal had no significant effect on the expression of *CSY* (F_1,20_ = 0.687, *p* = 0.417. Figure 5C). Temperature had no significant effect on *ACACβ* (F_1,20_ = 3.080, *p* = 0.095), but iBAT removal stimulated *ACACβ* expression significantly, particularly under cold condition (F_1,20_ = 11.197, *p* = 0.003. Figure 5D). The expression of *COX II* was significantly affected by temperature (F_1,20_ = 5.006, *p* = 0.037), but not by iBAT removal (F_1,20_ = 2.617, *p* = 0.121. Figure 5E), and multivariate analysis showed that *COX II* was higher in CR group than in WR and WS groups. The results showed that cold acclimation and iBAT removal enhanced mitochondrial metabolism, which indicates that thermogenic capacity was enhanced in muscle.

## 3. Discussion

Wild small rodents are natural endotherm species. When faced with natural seasonal environmental temperature fluctuations, compared with ectotherms such as nematodes, zebrafish, and model animal, such as rats and mice, wild animals’ response to environmental temperature changes is more similar to that of human beings. Therefore, it has more reference value for the study of human thermogenesis and thermoregulation. In addition, using wild animals instead of model animals can highlight the adaptability of animals to the environment. While the suggestion that skeletal muscle also plays a role in NST during cold acclimation in fur seals (*Callorhinus ursinus*), dogs, rats and rabbits [24,25,26,27,28], it remains unclear for the contribution of skeletal muscle-based NST during cold adaption in wild small rodents.. There may be compensatory mechanism induced for thermogenesis to different extents [29]. The main purpose of this study was to determine whether the skeletal muscles of Brandt’s voles produce heat at low temperature and whether they can compensate for thermogenesis when BAT function is minimized. The reason why we chose to remove iBAT instead of *UCP1*^−/−^ is that conditional ablation of iBAT is superior to using an *UCP1*^−/−^ model, which was discovered to be highly heterogeneous in cold sensitivity for mice with different genetic backgrounds [30]. In addition, since Brandt’s voles do not have a mature *UCP1*^−/−^ model, we removed the interscapular BAT by surgery to minimize the BAT function. Moreover, there has been limited research addressing how loss of BAT activity influences skeletal muscle in rodents during cold acclimation, especially Ca^2+^ cycling and mitochondrial metabolism. Here, we demonstrate that iBAT, rather than other parts of BAT, had a significant effect on cold-induced thermogenesis. Cold acclimation can increase skeletal muscle NST and cause WAT browning. When BAT function is acutely minimized, skeletal muscle becomes the major site of NST during cold acclimation, and muscle thermogenesis depends on SER Ca^2+^ handling and mitochondrial remodeling. The muscle NST as a thermogenic mechanism must have been wired into the vertebrate body quite early in the evolution and might play even greater roles in birds (where BAT is absent) and larger mammals (where BAT becomes a minor component in adulthood) [30]. At the early stage of evolution, birds lost the UCP1 gene from their genomes, and completely lacked the BAT-like structure [31,32,33]. There are some mammalian clades, such as marsupials and monotremes, that lack BAT, but they are endothermic [34,35,36]. This would mean that NST in skeletal muscle is more ancient than NST in BAT. In large mammals, such as pigs, BAT is lacking after adulthood [37]. This may be due to the rapid decline in BAT levels as muscles gain the ability to provide heat shortly after birth. The production of UCP1 is a very energy consuming process [38], which is consistent with the result that the amount of added BAT is very small after BAT removal. For Brandt’s vole, after BAT was minimized, it could still survive. This evidence may imply that the minimization of BAT leading to muscle NST enhancement may have evolutionary significance. However, new research has recently shown that systemic endothermy is ancient and common in amniotes. Among extant birds and mammals, muscle NST was driven by similar biochemical processes, which strengthens the case for its plesiomorphy [39].

### 3.1. Effect of Cold and Removal on Metabolic Phenotype, Thermogenesis in BAT and WAT

In winter, many small mammals reduce their body’s demand for heat by losing weight [40] such as *Microtus oeconomus* [41]. However, the body weight of Brandt’s voles was stable after 4 weeks of cold acclimation in this study, which may be related to the fact that they were only treated with low temperature without limited food resources. Our results show that the increase in food intake of animals under low temperature environment may be the main reason for compensating for the energy consumption caused by low temperature environment [42]. It can also be said cold exposure could help maintain weight and compensate for overeating.

BAT is a highly specialized organ in small mammals, and it is a major site of NST at neonatal and adult stages [43,44]. In many species such as rats [45], mice [46], deer mice (*Peromyscus maniculatus*) [47], cold acclimation activated iBAT thermogenesis. However, whether other parts of BAT were related with thermogenesis was unknown. Our results showed that the thermogenic function of BAT was related to its location. IBAT contributed to heat production during cold acclimation, but other parts of BAT showed no effect on thermogenesis. Moreover, iBAT removal did not activate the thermogenesis in other sites of BAT. While there was no difference in UCP1 expression in four groups, PGC1α was different. Studies have shown that SLN activates mitochondrial biosynthesis through PGC1α signaling [48]. Therefore, it is possible that the increased expression of PGC1α did not promote the expression of UCP1, but acts with SLN to promote the increase of heat production in muscle. In iBAT, the total protein content and total UCP1 content at low temperature are significantly higher than that at room temperature, which can also indicate that more fat is used for heat production in cold environment, resulting in lower fat content and higher protein content. Therefore, the biological activity per milligram of tissue in iBAT at cold condition is much greater than that at warm condition. For BAT in other parts, cold did not lead to significant difference in protein content, which also indicated that low temperature did not lead to changes in thermogenesis, which was consistent with the results of measured thermogenesis indicators. The removal of iBAT led to an increase in protein content, but it was not reflected in specific thermogenic indicators, which may be related to other functions. The previous study showed that cafeteria-feeding led to weight gain in interscapular, thoracic, and perirenal BAT in rats, and NST increased with the increase of BAT mass [49]. No study was found concerning the effect of low temperature on thermogenesis of BAT in other sites. Therefore, the function of other parts of BAT should be further investigated. WAT is the major energy reserve and will be mobilized to supply energy metabolism during increased physiological demand, such as prolonged cold acclimation. In addition, inguinal WAT will undergo browning, with increased UCP1 expression to produce heat uncoupling from ATP production, when animals are stimulated by cold [50]. Loss of classic iBAT in mice enhanced WAT browning [51], our research has similar results, which means iBAT removal induced browning of WAT. These results emphasized the roles of iBAT and WAT in thermoregulation during cold acclimation in small mammals.

### 3.2. SER Ca^2+^ Handling Played a Central Role in Muscle NST

Previous studies have speculated that increased shivering is the basis for cold acclimation in *UCP1*^−/−^ mice [52]. But shivering is an acute reaction. Continued shivering during long term cold adaption can cause fatigue and injury, impair muscle function and metabolic capacity, and even threaten survival [1]. As a result, NST needs to be activated during a transition from shivering to the cold-acclimated state. Studies have shown that, even though when shivering is reduced with the administration of a sufficient dose curare, an agent that blocks acetylcholine receptors at neuromuscular junctions, iBAT-ablated mice can survive an acute cold (4 °C) challenge, which suggests that BAT-independent NST exists even in mice [16]. Our study showing drastic changes in skeletal muscle, especially no obvious change of thermogenesis in BAT of other parts, during prolonged cold acclimation in iBAT-ablated voles provides evidence for the existence of muscle-based NST. Uncoupled SERCA activity increases ATP hydrolysis and thermogenesis, thus contributing to muscle-based adaptive thermogenesis [53,54,55]. *SLN*^−/−^ mice exposed to acute cold (4 °C) were found that the animals were not able to maintain core body temperature (37 °C) [16]. In order to make the results more convincing, we wanted to obtain the expression level of SLN, but after a lot of efforts, there was still no result. We speculated that SLN antibody currently available may not work for some species and is very tricky. We found that *SLN* expression was increased significantly upon iBAT ablation and a preferential expression of the SERCA1, which suggests that SLN/SERCA interaction plays an important role in muscle NST. PGC-1α was also measured in skeletal muscle, and cold up-regulated expression, which may be affected by *SLN*. Both *PLB* and P-PLB are not affected by temperature and iBAT removal. This study shows that PLB does not function like SLN. Studies in skeletal muscles in birds have shown that cold acclimation increases SER Ca^2+^ cycling and RyR1 content [56,57]. It has been shown that phosphorylation of RyR leads to enhanced Ca^2+^ leaking [58,59,60]. There is also an increase in RyR1 expression in iBAT-ablated voles in our study. Our data show similar mechanisms in skeletal muscles in Brandt’s voles, which indicate that the molecular mechanisms of muscle-based NST are analogous in both birds and mammals. Studies have shown that, in mammals, skeletal muscle fibers can change their heat production level by changing RyR1 Ca^2+^ leakage under resting state [61]. In “CR” group, *RyR1* was significantly upregulated. That is, RyR1 Ca^2+^ leakage rate is increased, which leads to the amplification of the basic ATP turnover rate of sarcoplasmic reticulum Ca^2+^ pumps. Results show that uncoupling of SERCA activity by SLN and RyR1 may play an important role in the muscle NST of Brandt’s voles.

Mitochondria are highly dynamic organelles and undergo fusion and fission continually in response to energy demand [62]. In muscle, high energy requirements activate mitochondrial fusion, which leads to increased proton motive force, so as to promote the production of ATP [63,64]. When rats were exposed to cold environment, expression of *UCP3* in skeletal muscles was rapidly up-regulated, and the expression level reached its peak one day later, however, long-term cold acclimation reduced the expression [65]. This indicates that UCP3 does not play the main role of adaptive thermogenesis because the increased NST lasts long days during cold acclimation. Our results that both the mRNA expression and protein levels of UCP3 in muscle did not change with different treatment conditions during long-term acclimation also support that UCP3 does not contribute to NST. Due to the change of acclimation conditions, mitochondrial machinery was substantially required for remodeling, including mitochondrial abundance and architecture. IBAT removal significantly increased the number of mitochondria in skeletal muscle at 4 °C. As the images were always less favorable, we only focused on the number of mitochondria, not on cristae density. Another study found that cristae density is increased in cold-adapted mice [66]. In the future, we need to improve the procedure of transmission electron microscopy to obtain more convincing results. At the same time, *citrate synthetase*, which is the rate-limiting-enzyme of the TCA cycle, *ACACβ* that catalyzes fatty acid oxidation, and *COX II* that participates in respiratory oxidation were all increased in expression under the influence of low temperature and iBAT removal, which provides evidence of increased energy generation.

## 4. Material and Methods

### 4.1. Animals

Brandt’s voles used in the experiments were from laboratory colonies in the Institute of Zoology, Chinese Academy of Sciences (CAS) in Beijing. The voles were housed with the same sex siblings in plastic cages (30 × 15 × 20 cm^3^) at a light regime of 16 h light:8 h dark (lights on from 4:00 to 20:00) and room temperature of 23 ± 1 °C. The voles were fed a standard rabbit pellet chow (containing 18% protein, 3% fat, 12% fiber, and 47% carbohydrate, Beijing Keao Xieli Feed Co., Ltd., Beijing, China) and were provided with water ad libitum. The animal procedures were approved by the Animal Care and Use Committee of Institute of Zoology, CAS.

### 4.2. Experimental Design

IBAT constitutes > 50% of the total BAT depots in Brandt’s voles. We surgically removed iBAT to minimize the contribution from BAT to reveal the importance of skeletal muscle in NST. A total of 24 adult male voles (aged 6 months) were housed singly and were acclimated to the cage for 2 weeks. Then 12 voles received iBAT resection, while the other 12 voles received corresponding sham surgery as controls. These voles were divided into 4 treatments: Cold (at 4 ± 1 °C) removal (CR, 6 voles), cold sham (CS, 6 voles), warm (at 23 ± 1 °C) removal (WR, 6 voles), and warm sham (WS, 6 voles). Four groups were acclimated for 28 days.

### 4.3. Surgical Removal of iBAT

Before surgery, the surgical instruments were sterilized by immersing in 75% alcohol for 30 min. The voles were anesthetized with 2% pentobarbital sodium (Sigma-Aldrich, Darmstadt, Germany) at a dose of 30 mg/kg through intraperitoneal injection. After being sterilized at the skin of the vole scapula with iodophor, the skin was cut along the dorsal line, and the length of the incision is about 1–2 cm. Then the white fat covered on the iBAT was peeled off, and iBAT was stripped away from surrounding muscle and connective tissue and was weighted (accurate to 0.001 g). In the sham surgery group, iBAT was only dissected from muscle, connective and adipose tissue without damaging the blood vessels and nerves. The skin was sutured with absorbable PGA sutures (Shanghai Jinhuan, Model R413, 4-0, Shanghai, China), and then disinfected again with iodophor. Upon awakening, the animals were put back to the animal room and recovered from the operation. The temperature acclimation began 6 days after surgery.

### 4.4. Body Weight and Energy Intake

Both body weight and food intake were measured in every three days. Voles were weighed at the beginning (day 0) and the end (day 27) of the acclimation using an electronic balance (±0.1 g). For food intake measurement, voles were placed in a cage with a known amount of food (~50 g). After 2 days, the food residues were collected and weighed, and food intake was calculated as the difference between these values.

### 4.5. Metabolic Measurements

To determine whether iBAT removal affected the metabolic state of the voles, we measured resting metabolic rates (RMR), which represents the minimal amount of energy expenditure while animals are at rest and within the thermoneutral zone [67,68]. We also measured NST, which is an important, yet energetically costly mechanism for small mammals to increase heat production and maintain stable body temperatures [68]. At the beginning of week 4, RMR values were measured as oxygen consumption using an open-circuit respirometry system (TSE labmaster, Thuringia, Germany). RMR were measured at 30 °C using 2.7 L metabolic chambers (Type I for mice), and the air flow rate was set at 0.8 L/min. Oxygen consumption was measured every 6 min for 3 h. We took the average of the 3 lowest consecutive readings as the RMR. The measurements of NST were conducted at 25 °C. We injected animals subcutaneously with norepinephrine (NE) (51-41-2, Shanghai Macklin Biochemical Co., Ltd., Shanghai, China) to elicit an elevated metabolic rate. The dosage of NE was calculated by the formula NE (mg/kg) = 2.53 BM^−0.4^ (BM, body mass) for Brandt’s voles [69]. Voles were then placed in the respirometry equipment described above, and the cost of NST is evaluated as the 3 highest consecutive readings of oxygen consumption (VO_2_) during 1 h of measurement.

### 4.6. Body Temperature

In the last week of acclimation, rectal temperature was measured using a digital thermometer (UT320, Uni-trend Technology Co., Ltd., Dongguan, China, ± 0.1 °C). The probe of the thermometer was inserted 3 cm into the rectum and the highest stable temperature reading was taken within 30 s. To determine whether the body temperature pattern among groups was stable, we measured body temperature at the end of each metabolic trial.

### 4.7. Tissue Collection

Prior to dissection, the hair was shaved off one of the animal’s hind legs, and the depilated leg was soaked in 4% paraformaldehyde (PFA) for more than 10 s. Animals were then sacrificed using CO_2_. Quickly the leg was removed and was soaked in 4% PFA. The muscle tissue was taken with the size of rice grain from the analyses of tibialis anterior (TA) muscle, and was soaked in the marked 1.5 mL eppendorf tube (4% PFA was added into the tube in advance), then was fixed with 2.5% glutaraldehyde solution (Sigma-Aldrich, Darmstadt, Germany) and stayed overnight. The gastrocnemius muscle was dissected from the other leg, quickly frozen in liquid nitrogen and stored at −80 °C for later protein and mRNA measurement. The inguinal WAT was taken and soaked in 4% PFA and fix for 24 h, or was frozen in liquid nitrogen. After routine paraffin embedding, a paraffin microtome was used to cut into 5 μm sections continuously, and 6–8 sections were placed horizontally on glass slides. We dissected BAT, including interscapular, infrascapular region, neck, and axilla, and measured the mass of each part, respectively.

### 4.8. Real-Time Quantitative PCR (RT-qPCR) for Measurements of Thermogenic and Metabolic Markers

We extracted total RNA from the skeletal muscles using TRIzol reagent (R401-01, Vazyme, Nanjing, China), and then 1 μg total RNA was purified and reverse-transcribed to cDNA using the HiScript^®^ III 1st Strand cDNA Synthesis Kit (+gDNA wiper) (R312-01/02, Vazyme, Nanjing, China). RT-qPCR analysis was carried out as follows: the cDNA samples (1 μL) were used as a template for the subsequent PCR reaction using gene-specific primers (Table 1). The final reaction volume of 10 μL contained 5 μL of 2×Taq Pro Universal SYBR qPCR Master mix (Q712-02, Vazyme, Nanjing, China), 1 μL cDNA template, 0.2 μL of forward primer, 0.2 μL reverse primer, and 3.6 μL RNase free ddH_2_O. Each sample was duplicated and the mean value was the expression amount of the sample. RT-qPCR was performed using Piko Real Software 2.2 (Piko Real 96, Thermo Scientific, Waltham, Massachusetts, MA, America). After an initial polymerase activation step at 95 °C for 30 s, amplification was followed by 40 cycles (95 °C for 10 s and 60 °C for 30 s). The reaction was finished by the built-in melting curve. All samples were quantified for relative quantity of gene expression by using *GAPDH* expression as an internal standard. Relative gene expression was determined by the comparative CT method [70].

### 4.9. Western Blotting (WB) for Measurements of Thermogenic Markers

Tissue samples of BAT (iBAT in two sham groups, and other parts of BAT in four groups) and skeletal muscle were weighed and homogenized in 200 µL of radioimmunoprecipitation assay buffer (RIPA buffer: 10 mM Tris, 158 mM NaCl, 1% TritonX-100, 5 mM EDTA, 1 mM DTT, 1 mM PMSF, and 1:1000 parts of protease inhibitor cocktail [P8340, Sigma-Aldrich, Darmstadt, Germany]) [71]. Homogenates were centrifuged at 13,000× g at 4 °C for 30 min. Protein from the supernatant was placed in 5×SDS-PAGE protein loading buffer (Yeasen, Shanghai, China) and denatured by heating at 100 °C for 5 min. Total loading protein was 40 μg and was separated by SDS-PAGE using a Mini Protean apparatus (BioRad Laboratories, Hercules, State of California, America) then transferred to polyvinylidene fluoride membranes. Membranes were incubated for 2 h at room temperature in 5% skim milk powder to reduce nonspecific antibody binding. The membranes were then exposed to primary antibodies (Table 2) for >12 h at 4 °C [71]. Whole tissue lysates of other parts of BAT were used to measure the expression of PCNA, UCP1, PGC-1α, PPARγ, DIO2, TH. Whole tissue lysates of iBAT were used to measure the expression of UCP1, PGC-1α, PPARγ, DIO2, and TH. Additinoally, whole tissue lysates of skeletal muscles were used to measure the expression of SERCA1, P-PLB and PGC-1α. β-tubulin or GAPDH was used as a housekeeping protein. Then, membranes were exposed to appropriate secondary antibodies for 2 h at room temperature (either peroxidase-conjugated goat anti-rabbit IgG (33101ES60, Yeasen, Shanghai, China) or peroxidase-conjugated goat anti-mice IgG (33201ES60, Yeasen, Shanghai, China) depending on the primary antibody). Reaction products were visualized by chemiluminescence (ECL, Yeasen, Shanghai, China). Protein was quantified with Image J, expressed as relative units to housekeeping proteins.

### 4.10. Immunohistochemistry for Detecting Browning of WAT

WAT sections were dewaxed with xylene and then rehydrated with gradient alcohol. Sections were incubated in 0.1 M PBS (P1003, Solarbio, Beijing, China) for 15 min, then denatured by boiling in Citric acid antigen repair buffer (AR-5011, Beijing Dingguo Changsheng Biotechnology Co., Ltd., Beijing, China) for 30 min. After 1 h of incubation with 10% normal goat serum (NGS, S9070, Solarbio, Beijing, China) containing 0.5% Triton-X 100 in 0.1 M PBS, sections were stained using anti-UCP1 antibody (ab10983, Abcam, Cambridge, England) at 1:1000 dilution for >12 h at 4 °C. Biotin conjugated goat-anti-rabbit (1:600, 33101ES60, Yeasen, Shanghai, China) were used as second antibodies at room temperature for 2 h. Sections were placed into ABC complex (PK-6100, Vector Labs, San Francisco, State of California, America) for 90 min, and positive cells were visualized with DAB (SK-4100, Vector Labs, San Francisco, State of California, America). The negative controls processing the secondary antibody with the omission of the primary were performed to verify the specificity of the label.

### 4.11. Transmission Electron Microscopy for Visualizing Mitochondria of Muscle Tissues

To visualize the mitochondrial ultrastructure of skeletal muscle in Brandt’s voles, we used transmission electron microscopy (JEM-1400, JEOL, Tokyo, Japan). First, muscle tissues were removed from 4% PFA and put in the 2.5% glutaraldehyde. The tissue material of approximately 1 mm^3^ was cut using a double-sided blade, followed by fine fixation in 1% osmium tetroxide for 1.5 h at 4 °C. Then, tissues were dehydrated and embedded in LR White resin (Electron Microscopy Sciences, Hatfield, England). Thin sections (80 nm) were prepared, stained with uranyl acetate and lead citrate and examined using a JEM-1400 transmission electron microscope.

### 4.12. Statistical Analysis 

We used the software SPSS 26.0 for statistical analyses. Body mass and food intake during the acclimation were analyzed by repeated measures ANOVA or ANCOVA. Differences in RMR and NST between groups were analyzed by two-way ANCOVA with body mass as covariate. The rectal temperature, mRNA expression and protein levels of thermogenic markers, and UCP1 positive ratio were analyzed by independent *t*-test, two-way or multivariate ANOVA with Tukey’s post hoc tests. The results are presented as means  ±  SEM, and the level of statistical significance was set at *p*  <  0.05. The figures in this article were made by GraphPad Prism 8.0.1.

## 5. Conclusions

The current study presents a new insight into the contribution of skeletal muscles to NST in wild rodents. It is iBAT, but not other parts of BAT, that contributes to NST, and when the voles experience cold, WAT enhances browning with increased UCP1 levels. Under cold acclimation, skeletal muscles enhance thermogenesis by increasing the mitochondrial number and mitochondrial metabolism, and up-regulating levels of major Ca^2+^-cycling proteins. IBAT removal improves the browning degree of WAT and *RYR1* expression in muscle to a greater extent, but it has no effect on thermogenesis-related targets in other sites of BAT (Figure 6). Our results suggest that iBAT and skeletal muscle play vital roles in cold-induced thermogenesis and skeletal muscle could compensate for impairment in BAT thermogenesis. It also implies that interaction in the thermogenic function of BAT and muscle helps small mammals to cope with cold stress for survival, especially in winter. If only one kind of tissue has the function, it can be life-threatening when it is damaged, and therefore, the interaction between the two forms of thermogenesis is more conducive to the survival and reproduction of small mammal species.

## Figures and Tables

**Figure 1 ijms-24-00342-f001:**
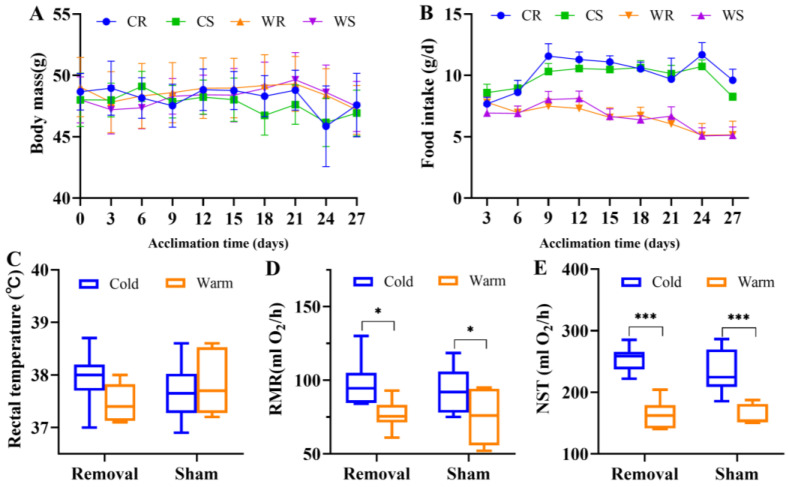
Interscapular brown adipose tissue (iBAT) removal did not induce alterations in metabolic phenotypes. Body mass (**A**), food intake (**B**), rectal temperature (**C**), resting metabolic rates (RMR, (**D**)), and nonshivering thermogenesis (NST, (**E**)). Values are means ± SEM. *, *p* < 0.05, ***, *p* < 0.001. CR, cold removal group. CS, cold sham group. WR, warm removal group. WS, warm sham group.

**Figure 2 ijms-24-00342-f002:**
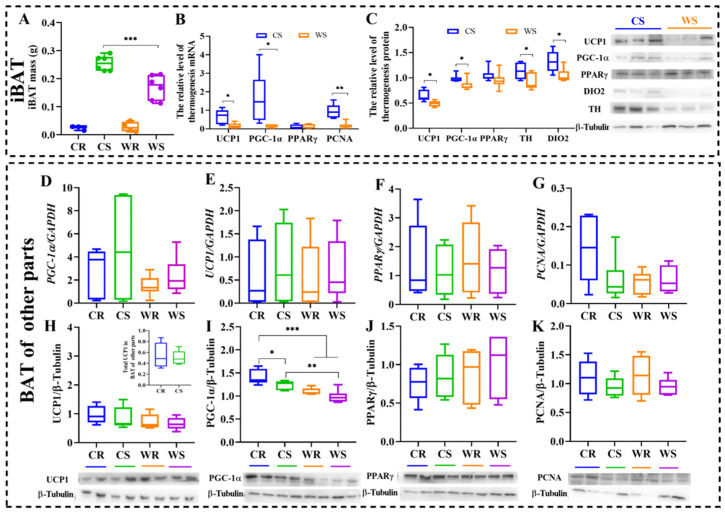
Cold and iBAT removal did not stimulate heat production in other sites of BAT. Interscapular brown adipose tissue (iBAT) mass (**A**), mRNA expression of *uncoupling protein 1*(*UCP1*), *peroxisome proliferator-activated receptor gamma coactivator 1-alpha* (*PGC-1α*), *peroxisome proliferator-activated receptor gamma* (*PPARγ*), and *proliferating cell nuclear antigen* (*PCNA)* in iBAT (**B**), protein levels of UCP1, PGC-1α, PPARγ, deiodinase iodothyronine type II (DIO2), and tyrosine hydroxylase (TH) in iBAT (**C**), mRNA expression of *UCP1*, *PGC-1α*, *PPRAγ*, and *PCNA* in other parts (infrascapular region, neck, axilla) of BAT (**D**–**G**), and protein levels of UCP1, PGC-1α, PPRAγ, and PCNA in other sites of BAT (**H**–**K**). The inserted panel in (**H**) shows total UCP1 in BAT of other parts. Values are means ± SEM. *, *p* < 0.05, **, *p* < 0.01, ***, *p* < 0.001. CR, cold removal group. CS, cold sham group. WR, warm removal group. WS, warm sham group.

**Figure 3 ijms-24-00342-f003:**
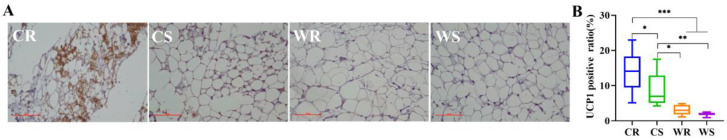
Cold and iBAT removal induced browning of WAT. The browning area of white fat in cold removal (CR), cold sham (CS), warm removal (WR) and warm sham (WS) group, 200× (**A**), scale bar = 100 μm. Expression of UCP1 in white adipose tissue (WAT, (**B**)). Values are means ± SEM. *, *p* < 0.05, **, *p* < 0.01, ***, *p* < 0.001.

**Figure 4 ijms-24-00342-f004:**
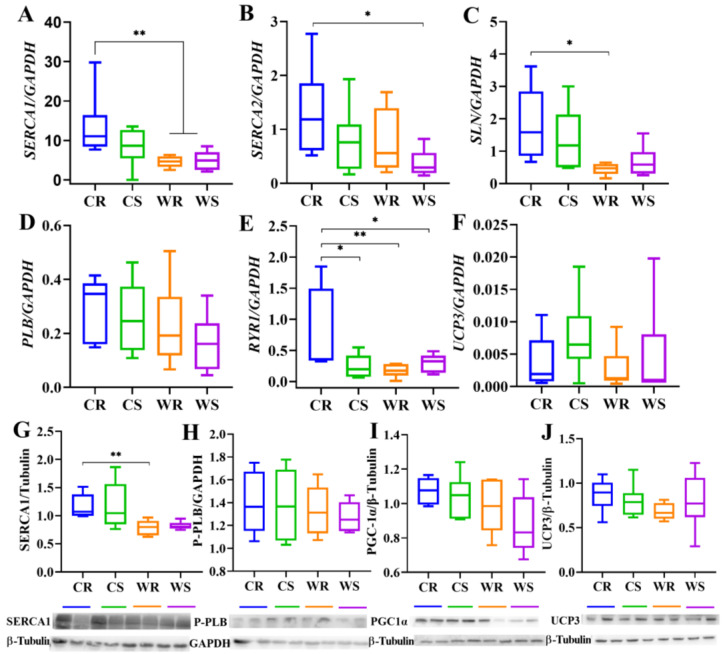
The mRNA expression and protein levels of thermogenesis-related markers in skeletal muscle. The mRNA expression of *sarcoplasmic endoplasmic reticulum Ca^2+^-dependent adenosine triphosphatase 1* (*SERCA1*), *sarcoplasmic endoplasmic reticulum Ca^2+^-dependent adenosine triphosphatase 2* (*SERCA2*), *sarcolipin* (*SLN*), *phospholamban* (*PLB*), *type 1 ryanodine receptor* (*RYR1*) and *uncoupling protein 3* (*UCP3*) (**A**–**F**), and protein levels of SERCA1,phosphorylated-PLB (P-PLB), peroxisome proliferator-activated receptor gamma coactivator 1-alpha (PGC-1α) and uncoupling protein 3 (UCP3) (**G**–**J**). Values are means ± SEM. *, *p* < 0.05, **, *p* < 0.01. CR, cold removal group. CS, cold sham group. WR, warm removal group. WS, warm sham group.

**Figure 5 ijms-24-00342-f005:**
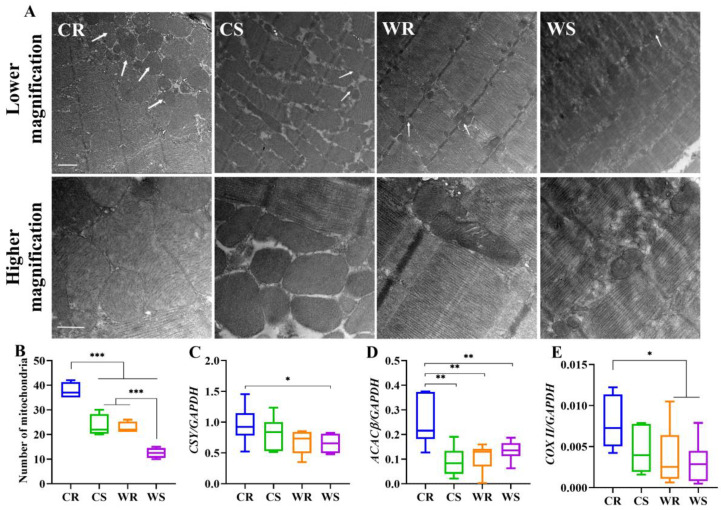
Cold and iBAT removal induce alterations in mitochondrial architecture and metabolism-related genes. Low magnification electron micrographs of tibialis anterior (TA) muscle, 5000×, and high magnification electron micrographs of TA, 14,000× (**A**). The white arrow points to mitochondria. Scale bar =1 μm in lower magnification and 500 nm in higher magnification. Number of mitochondria in each treatment (**B**). The mRNA expression of *citrate synthetase* (*CSY*, **C**), *acetyl-CoA carboxylase beta* (*ACACβ*, **D**), *cytochrome C oxidase subunit II* (*COX II*, (**E**)). Values are means ± SEM. *, *p* < 0.05, **, *p* < 0.01,***, *p* < 0.001. CR, cold removal group. CS, cold sham group. WR, warm removal group. WS, warm sham group.

**Figure 6 ijms-24-00342-f006:**
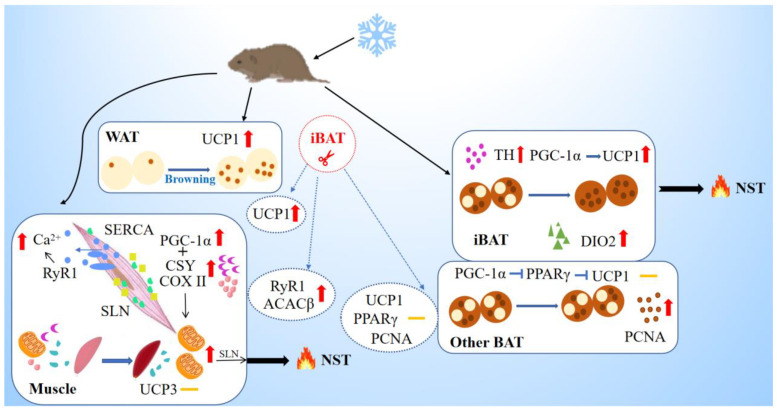
Schematic model of nonshivering thermogenesis induced by cold and iBAT removal in Brandt’s voles. The box represents cold-induced thermogenesis of tissues, and the circle represents the effect of iBAT removal. Yellow dashes represent no significant change in indicators. Cold acclimation induces browning of WAT, and results in up-regulation in the levels of TH, PGC-1α, UCP1 and DIO2 for thermogenesis in iBAT, but does not affect BAT in other sites except for increased PCNA level. Upregulation of PGC1a would promote mitochondrial biogenesis in the muscle (assisted by *COXII* & *CSY*) that will support *SLN*-mediated thermogenesis. Upregulated *RyR1* will increase leak and cytosolic Ca^2+^ that will activate muscle NST. Cold enhanced thermogenesis indicated by increased expression of *SERCA,* but with no change in UCP3. IBAT removal induced supplement in WAT browning and muscle metabolism and thermogenesis indicated by increased expression of *ACACβ* and *RYR1*. WAT, white adipose tissue; iBAT, interscapular brown adipose tissue; UCP1, uncoupling protein 1; TH, tyrosine hydroxylase; PGC-1α, peroxisome proliferator-activated receptor gamma coactivator 1-alpha; PPARγ, peroxisome proliferator-activated receptor gamma; DIO2, deiodinase iodothyronine type II; PCNA, proliferating cell nuclear antigen; CSY, citrate synthetase; ACACβ, acetyl-CoA carboxylase beta; COX II, cytochrome C oxidase subunit II; SERCA, sarcoplasmic endoplasmic reticulum Ca^2+^-dependent adenosine triphosphatase; SLN, sarcolipin; RyR1, the type 1 ryanodine receptor; UCP3, uncoupling protein 3; NST, nonshivering thermogenesis.

**Table 1 ijms-24-00342-t001:** The gene-specific primer sequences used for Real-time -qPCR.

Primers	Forward Primer (5′-3′)	Reverse Primer (5′-3′)
*SERCA1*	GATCCGAGACCAGATGGCTG	CAGGGTCGTTGAAGTGACCA
*SERCA2*	GTCTGTCATTCGGGAGTGGG	GCTGAGTCTTCCAGGTGCAT
*SLN*	GTCTGCCTGGAGTTCTCACC	ACGGCCCCTCAGTATTGGTA
*RYR1*	TGGTGGGCGAAATCTTCATCT	GGTCTGAGCCACCTGACTTG
*PLB*	AGCAAGCACGGCAAAATCTC	GGTGGCAGCCGTACTTCATA
*UCP3*	GCACAGTTGACAATGGCGTT	CTGCCTGAACTTGGCCCATA
*CSY*	GCTAAGGGTGGGGAAGAACC	ACCACATGAGAAGGCAGAGC
*ACACβ*	TCAACACAGCCTACGTCACC	GGGTACTTTTCTGGGGAGCC
*COX II*	CCCATAGAGCTCCCAATCCG	GTCGTCCTGGGATAGCATCTG
*GAPDH*	TGCTCCTCCCTGTTTTGGAG	TCCAATACGGCCAAATCCGT

**Table 2 ijms-24-00342-t002:** Antibodies for Western blotting.

Primary Antibody	Host	Antibody Type	Article Number	Manufacturer
PCNA (PC10) Mouse mAb	Mouse	mAb	2586	Cell Signaling Technology
anti-UCP1 antibody	Rabbit	pAb	ab10983	Abcam
PPARγ (C26H12) Rabbit mAb	Rabbit	mAb	2435	Cell Signaling Technology
Rabbit Anti-PPARGC1A (PGC-1α) Polyclonal Antibody	Rabbit	pAb	bs-1832R	Bioss
Polyclonal Antibody to DIO2	Rabbit	pAb	PAC903Ra01	Cloud-Clone Corp.
Anti-Tyrosine Hydroxylase Antibody	Rabbit	pAb	AB152	Merck Millipore
ATP2A1/SERCA1 (L24) Antibody	Rabbit	pAb	4219	Cell Signaling Technology
Phospho-Phospholamban (Ser16/Thr17) Antibody	Rabbit	pAb	8496	Cell Signaling Technology
β-Tubulin Mouse mAb	Mouse	mAb	30301ES40	Yeasen
GAPDH (Clone:1A6) Mouse mAb	Mouse	mAb	30201ES20	Yeasen

## Data Availability

The data presented in this study are available on request from the corresponding author.

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
