# Peer review of "Recruitment of Muscle Genes as an Effect of Brown Adipose Tissue Ablation in Cold-Acclimated Brandt’s Voles (Lasiopodomys brandtii)"

_ijms, 2022, doi:10.3390/ijms24010342_

Round 1
Reviewer 1 Report
The manuscript by Liu and colleagues concerns the degree to which brown adipose tissue thermogenesis is an essential component in nonshivering thermogenesis (NST) following cold acclimation in Brandt’s voles. While the question is of interest, the authors have not, in my opinion, adequately addressed it in the present version of the manuscript.
The question of which tissues participate in NST has been controversial since the discovery of the phenomenon in the 1950’s. Most workers would today subscribe to the idea that brown adipose tissue (BAT) and its mitochondrial protein UCP1 are the only or main players. This is thanks to paradigm-changing work by David Foster and later work with UCP1-knock-out mice. However, more recently, the old ideas of some component of NST deriving from skeletal muscle have been re-activated. To date, the evidence, when strictly analyzed, is not convincing but the idea is apparently attractive. Changes in different proteins that influence cytosolic calcium levels are forwarded as evidence but it is difficult presently to exclude e.g. that such changes arise from sustained (micro)shivering and the use of too low doses of curare preclude conclusions of an absence of muscle shivering.
It is important to emphasize that in the present manuscript, the authors have not measured muscle thermogenesis but only norepinephrine-induced thermogenesis. Therefore, it is essential that the title of the manuscript is changed to reflect this. I would suggest: Recruitment of muscle genes as an effect of brown adipose tissue ablation in cold-acclimated Brandt’s voles.
The conclusions of the authors are thus primarily indirect and can be interpreted in different ways. Also, their measurements of gene expression, either as mRNA or as protein, are conventional and therefore do not provide a physiological perspective on the changes. The authors have removed ≈ 50 % of the BAT and subsequently cold-acclimated the animals. At the conclusion of the experiment, they removed the relevant tissues quantitatively for analysis. They performed qPCR and Western blots. Thus, their results are relative and not absolute.
The animals, iBAT-ablated or not, are clearly able to cope with the cold temperatures, as indicated by the fact that ablated and non-ablated animals do not differ in body mass, food intake or rectal temperature. As expected, RMR and NE-induced respiration are elevated following cold acclimation and are not different between ablated and non-ablated. This indicates that some form of compensation has occurred subsequent to the ablation of iBAT.
One point the authors do not comment but which will have consequences for interpretation is that while the weight of iBAT is given, its composition is not indicated. Clearly, in the warm, the tissue will be fat-filled, while in the cold, as a result of increased cell proliferation and mitochondriogenesis, as well as utilization of lipids for heat, the tissue will have a much lower fat content and higher protein content. Thus, its biological activity will be much greater in the cold-acclimated tissue per mg tissue. Since the authors clearly have the weights of the tissue and the protein density, they can calculate the total protein content per depot. This can then be used, see below, to obtain a physiological perspective on their results.
The increase in UCP1 expression in iBAT of non-ablated animals is fairly low, probably because the control animals were maintained below thermoneutrality and thus already had elevated their UCP1 expression. However, if the authors use the arbitrary values of UCP1 expression per mg protein and then multiply this by the protein amount in the total tissue depot, they will obtain an estimate of the total UCP1 in that depot. This is then the physiologically relevant value.
This way to estimate UCP1 content should then be repeated for all the BAT depots that the authors have quantitatively dissected from all the animals, both ablated and non-ablated. When the values for the different depots are then summed, this will provide more information as to whether compensation has in fact occurred in the summed adipose depots (including the inguinal, which should be analyzed in the same way as the brown depots). The present conclusion that compensation has not occurred is only based on relative values and may thus be misleading.
The authors should also bear in mind that the histology as seen in Fig. 3 is difficult to interpret because there are obviously very different numbers of cells per field because of the changes in lipid content. While I do not disagree that browning has occurred, if the authors wish to use this approach, they should count the same number of cells and not per field.
The authors are enthusiastic about the changes in skeletal muscle in the BAT-ablated animals. However, it is clear that the changes overall are modest. Part of the problem is that only four animals are included per group and the standard errors are very large, causing difficulty in interpretation. As I indicated above, the changes could possibly reflect enhanced shivering ability. The muscle data should then be discussed in relation to the recalculated BAT data.
Author Response
The manuscript by Liu and colleagues concerns the degree to which brown adipose tissue thermogenesis is an essential component in nonshivering thermogenesis (NST) following cold acclimation in Brandt’s voles. While the question is of interest, the authors have not, in my opinion, adequately addressed it in the present version of the manuscript.
Reply: At present, skeletal muscle non shivering thermogenesis is mainly concentrated in birds and large mammals. For small mammals, the research is also about model animals, such as mice. small wild mammals, which have evolved an obligate thermogenic organ of BAT, it remains unclear whether non-shivering effects of skeletal muscle really exist, and whether skeletal muscle and BAT can compensate for each other in terms of heat production. Based on this situation, we propose two scientific questions, whether the skeletal muscles of Brandt’s voles produce heat at low temperature and whether they can compensate for thermogenesis when BAT function is minimized. Under cold acclimation, skeletal muscles enhance thermogenesis by increasing mitochondrial number and mitochondrial metabolism, and up-regulating levels of major Ca2+-cycling proteins. Our results suggest that iBAT and skeletal muscle play vital roles in cold-induced thermogenesis and skeletal muscle could compensate for impairment in BAT thermogenesis.
The question of which tissues participate in NST has been controversial since the discovery of the phenomenon in the 1950’s. Most workers would today subscribe to the idea that brown adipose tissue (BAT) and its mitochondrial protein UCP1 are the only or main players. This is thanks to paradigm-changing work by David Foster and later work with UCP1-knock-out mice. However, more recently, the old ideas of some component of NST deriving from skeletal muscle have been re-activated. To date, the evidence, when strictly analyzed, is not convincing but the idea is apparently attractive. Changes in different proteins that influence cytosolic calcium levels are forwarded as evidence but it is difficult presently to exclude e.g. that such changes arise from sustained (micro)shivering and the use of too low doses of curare preclude conclusions of an absence of muscle shivering.
Reply: At present, we really can not completely rule out the shivering thermogenesis (SH) of skeletal muscle. But we can explain it laterally. Shivering is an acute response and can last only few seconds at a time. Continued shivering during prolonged cold adaptation is detrimental for muscle function, as it can cause fatigue and injury and weaken muscle function and metabolic capacity. Our temperature acclimation is a continuous process, rather than an instantaneous process from room temperature to cold environment. Therefore, I think that even if there is shivering thermogenesis, the effect of shivering thermogenesis may be insignificant compared to that of non shivering thermogenesis (NST) for a long-term low temperature environment. In the long-term keep out the cold process, only under severe cold exposure conditions, SH can be activated as an additional heat source. For example, studies have shown that, in winter, Phodopus sungorus produce heat through NST in the whole ambient temperature range from 20 to - 36℃, SH gradually increased only when the ambient temperature was below -36℃[1] (Figure 1). There are also studies that have found that iBAT-ablated mice can survive an acute cold (4 ℃) challenge even when shivering is minimized by administration of curare (an agent that blocks acetylcholine receptors at neuromuscular junctions)[2], suggesting the existence of BAT-independent NST even in mice.
Figure 1
[1] Heldmaier G, Böckler H, Buchberger A, Lynch G, Puchalski W, Steinlechner S, Wiesinger H.
Seasonal acclimation and thermogenesis, Circulation, Respiration, and Metabolism. Circulation, Respiration, and Metabolism. Springer, Berlin, Heidelberg, 1985: 490-501.
[2] Bal N. C., Maurya S. K., Sopariwala D. H., Sahoo S. K., Gupta S. C., Shaikh S. A., Pant M., Rowland L. A., Bombardier E., Goonasekera S. A., Tupling A. R., Molkentin J. D., and Periasamy M. Sarcolipin is a newly identified regulator of muscle-based thermogenesis in mammals. Nat. Med. 2012, 18, 1575–1579
It is important to emphasize that in the present manuscript, the authors have not measured muscle thermogenesis but only norepinephrine-induced thermogenesis. Therefore, it is essential that the title of the manuscript is changed to reflect this. I would suggest: Recruitment of muscle genes as an effect of brown adipose tissue ablation in cold-acclimated Brandt’s voles.
Reply: Thank you for your suggestion. It is true that we only measured the expression of genes and proteins related to muscle thermogenesis but did not directly measure muscle thermogenesis. Therefore, the original title is not very appropriate, and we have changed it to “Recruitment of Muscle Genes as an Effect of Brown Adipose Tissue Ablation in Cold-acclimated in Brandt's Voles (Lasiopodomys brandtii)”.
The conclusions of the authors are thus primarily indirect and can be interpreted in different ways. Also, their measurements of gene expression, either as mRNA or as protein, are conventional and therefore do not provide a physiological perspective on the changes. The authors have removed ≈ 50 % of the BAT and subsequently cold-acclimated the animals. At the conclusion of the experiment, they removed the relevant tissues quantitatively for analysis. They performed qPCR and Western blots. Thus, their results are relative and not absolute.
Reply: We do not deny that the results of qPCR and Western are relative rather than absolute, because they are calculated by comparing the target gene or protein with the housekeeping gene or protein, rather than being directly measured like RMR and NST. It was found that the UCP1 mRNA of voles increased 2.74 times after cold acclimation[3]. And research showed that cold caused a significant increase of UCP1 mRNA in the hamsters[4], which was consistent with the enhancement of NST at individual levels[5]. So I think such results should also reflect the consistent relationship between mRNA or protein levels and dynamic physiology.
[3]Li Q, Sun R, Huang C, Wang Z, Liu X, Hou J, Liu J, Cai L, Li N, Zhang S, Wang Y. Cold adaptive thermogenesis in small mammals from different geographical zones of China. Comp Biochem Physiol A Mol Integr Physiol. 2001 Jul;129(4):949-61.
[4]von Praun C, Burkert M, Gessner M, Klingenspor M. Tissue-specific expression and cold-induced mRNA levels of uncoupling proteins in the Djungarian hamster. Physiol Biochem Zool. 2001 Mar-Apr;74(2):203-11.
[5]Wiesinger H, Heldmaier G, Buchberger A. Effect of photoperiod and acclimation temperature on nonshivering thermogenesis and GDP-binding of brown fat mitochondria in the Djungarian hamster Phodopus s. sungorus. Pflugers Arch. 1989 Apr;413(6):667-72.
The animals, iBAT-ablated or not, are clearly able to cope with the cold temperatures, as indicated by the fact that ablated and non-ablated animals do not differ in body mass, food intake or rectal temperature. As expected, RMR and NE-induced respiration are elevated following cold acclimation and are not different between ablated and non-ablated. This indicates that some form of compensation has occurred subsequent to the ablation of iBAT.
Reply: The results proved that this was true, after iBAT ablation, some compensatory effects occurred in other sites. Our results indicate that under the same temperature conditions, whether iBAT was removed or not was not significantly affect the basic metabolic phenotype. So we guess that the heat produced by the lost iBAT must be compensated by other parts. According to this conjecture, we selected several thermogenic sites, WAT, BAT in other sites and skeletal muscle to measure their thermogenic indicators to confirm our idea. This leads to some of the results in this article.
One point the authors do not comment but which will have consequences for interpretation is that while the weight of iBAT is given, its composition is not indicated. Clearly, in the warm, the tissue will be fat-filled, while in the cold, as a result of increased cell proliferation and mitochondriogenesis, as well as utilization of lipids for heat, the tissue will have a much lower fat content and higher protein content. Thus, its biological activity will be much greater in the cold-acclimated tissue per mg tissue. Since the authors clearly have the weights of the tissue and the protein density, they can calculate the total protein content per depot. This can then be used, see below, to obtain a physiological perspective on their results.
The increase in UCP1 expression in iBAT of non-ablated animals is fairly low, probably because the control animals were maintained below thermoneutrality and thus already had elevated their UCP1 expression. However, if the authors use the arbitrary values of UCP1 expression per mg protein and then multiply this by the protein amount in the total tissue depot, they will obtain an estimate of the total UCP1 in that depot. This is then the physiologically relevant value.
Reply: Thank you for your suggestion. Because total protein content and total UCP1 content are put together for discussion, the questions raised by the reviewer are answered together.
According to your suggestion, we have made three figures about protein content and put them in Figure 2 (Fig.2L, Fig.2M, Fig.2N). Including total protein content of BAT in other parts, total protein content of iBAT in sham groups and total UCP1 content of iBAT in sham groups. Corresponding contents have also been added in the result, legend and discussion.
The content added to the result is “Cold had no significant effect on the total protein content of BAT in other parts (F1,20=1.054, p=0.317), but iBAT removal had(F1,20=19.138, p<0.001. Fig.2L). Cold acclimation increased total protein content of iBAT (t=4.617, df=10, p=0.001. Fig.2M) and total UCP1content (t=3.146, df=10, p=0.027. Fig.2N). Cold resulted in significant increase of protein content in iBAT, but had no effect on BAT in other parts”.
The content added to the figure legend is “Total protein content of BAT in other parts (L), total protein content of iBAT in sham groups (M), total UCP1 content of iBAT in sham groups (N)”.
The content added to the discussion is “In iBAT, the total protein content and total UCP1 content at low temperature are significantly higher than that at room temperature, which can also indicate that more fat is used for heat production in cold environment, resulting in lower fat content and higher protein content. Therefore, the biological activity per milligram of tissue in iBAT at cold condition is much greater than that at warm condition. For BAT in other parts, cold did not lead to significant difference in protein content, which also indicated that low temperature did not lead to changes in thermogenesis, which was consistent with the results of measured thermogenesis indicators. The removal of iBAT led to an increase in protein content, but it was not reflected in specific thermogenic indicators, which may be related to other functions”.
This way to estimate UCP1 content should then be repeated for all the BAT depots that the authors have quantitatively dissected from all the animals, both ablated and non-ablated. When the values for the different depots are then summed, this will provide more information as to whether compensation has in fact occurred in the summed adipose depots (including the inguinal, which should be analyzed in the same way as the brown depots). The present conclusion that compensation has not occurred is only based on relative values and may thus be misleading.
Reply: Thank you for your suggestion. Many studies have proved that iBAT is the main part of thermogenesis in small mammals, and our research group has also proved this result. UCP1 is considered to be a key thermogenic protein. From the perspective of measuring thermogenesis, I think the supplementation of the total protein content and the total UCP1 content you suggested can be illustrative, so I added the figure of total UCP1 content instead of regulating proteins.
The authors should also bear in mind that the histology as seen in Fig. 3 is difficult to interpret because there are obviously very different numbers of cells per field because of the changes in lipid content. While I do not disagree that browning has occurred, if the authors wish to use this approach, they should count the same number of cells and not per field.
Reply: Thank you very much for your advice. I understand what you said and think it is correct. However, in some references[6,7], there are also such a representation method as mine, which uses the browning area under a visual field to represent the browning degree. Therefore, I think my result is also feasible.
- Bal NC, Singh S, Reis FCG, Maurya SK, Pani S, Rowland LA, Periasamy M. Both brown adipose tissue and skeletal muscle thermogenesis processes are activated during mild to severe cold adaptation in mice. J Biol Chem. 2017, 6;292(40):16616-16625.
- Bo TB. Regulation Mechanism of Gut Microbiota on Energy Metabolism of Brandt's Vole (Lasiopodomys brandtii). Phd dissertation. University of Chinese Academy of Sciences, Beijing. 2020: 33-34
The authors are enthusiastic about the changes in skeletal muscle in the BAT-ablated animals. However, it is clear that the changes overall are modest. Part of the problem is that only four animals are included per group and the standard errors are very large, causing difficulty in interpretation. As I indicated above, the changes could possibly reflect enhanced shivering ability. The muscle data should then be discussed in relation to the recalculated BAT data.
Reply: When taking samples, there were 6 animals in each group. The amount of repetition may be controversial. In the following experiments, we will increase the number of animals to make the data more convincing. With regard to the selection of indicators about NST in skeletal muscle, we referred to many literatures, finally determined Ca2+-cycling proteins, and measured the expression of mRNA and protein. We do not deny that these expression levels are relative rather than absolute, but many literatures also use such data as evidence of skeletal muscle thermogenesis, so we also choose such indicators to represent the thermogenesis, resulting in results that can only prove the correlation between skeletal muscle and NST, rather than absolute. Your suggestions are very valuable, which are very helpful for our future research. It makes us clear what we should do to get more convincing evidence.

Reviewer 2 Report
The manuscript presents an interesting topic and shows that BAT is not the only source of NST. In the absence of BAT, skeletal muscle can contribute to cold-induced thermogenesis and compensate for impaired BAT function in wild rodents through several mechanisms. This paper is well-written with clear background and aims. Clarifications concerning the implications of this research are required before considering this paper for publication. I have additionally highlighted some minor comments below.
Major comments:
1. You have indicated that this study has the advantage of using small wild rodents over mice. Could you clarify the reason? Are small wild rodents more metabolically similar to humans, hence it is important to study them?
2. Following on from point 1, what are the implications of this research? There doesn’t seem to be any mention of human studies (I presume this is the ultimate aim to extrapolate results to humans). This is a major point to consider.
3. An interesting point was the increased food intake among those exposed to cold temperature, yet that hasn’t been discussed. In the discussion, please elaborate on this point (cold exposure could help maintain weight and compensate for overeating).
4. It seems the NST temperature for Brandt’s voles is 4 degrees C. Explain how you have determined this threshold.
Minor comments:
Abstract: “which are lack of BAT”: please rephrase
Figure 1: Please explain in the legend what do “CR CS WR WS” refer to.
Discussion: please adjust superscripts & remove [Error! Bookmark not defined].
Round 2
Reviewer 1 Report
The manuscript has been improved. However, there are some points (that I already mentioned) that still should be improved before acceptance. 1) That shivering, as the authors write, should only be able to exist for a few seconds at a time is clearly incorrect and many birds shiver continuously (see Heldmaier). Thus, the further discussion that NST exists in birds is not adequately demonstrated and is controversial. Thus, this cannot be used to argue that muscle-NST exists in mammals. In reality, any observations on birds cannot be extrapolated to animals - and there is, despite what the authors imply, no convincing evidence for the existence of NST in birds. The text should reflect this. 2) The authors continue to quote the curare experiment in spite of me stating that the dose given was obviously inadequate to prevent muscle activity (which would include shivering). Animals given sufficient curare must be ventilated and are unable to move. The film connected to the article shows that this is not the case and therefore the experiment is invalid. It cannot be used as an argument in this case. That the paper was accepted in Nature does not validify incorrect experiments. The text references to this paper must at least include the information that an adequate dose of curare was not used. 3) The authors have attempted to calculate the total UCP1 in the “other” BAT depots but seem to have misunderstood the task. The important value is the total UCP1 amount in the CR animals versus the CS animals, not versus any animals in the warm. Please attempt again to recalculate. A total amount cannot be per gram, it must be per total tissue (amount per gram multiplied with tissue weight)
Author Response
- 正如作者所写,颤抖一次只能存在几秒钟显然是不正确的,许多鸟类不断颤抖(见Heldmaier)。因此,NST存在于鸟类中的进一步讨论没有得到充分证明,并且存在争议。因此,这不能用来论证肌肉NST存在于哺乳动物中。实际上,任何对鸟类的观察都不能外推到动物身上——尽管作者暗示,但没有令人信服的证据证明鸟类中存在NST。案文应反映这一点。
回复:对不起,我没有说清楚。我的初衷是,颤抖是一种急性反应。在长时间的寒冷适应期间持续颤抖对肌肉功能有害,因为它会导致疲劳和受伤,并削弱肌肉功能和代谢能力。长时间的颤抖可能代价高昂,并可能严重损害生物体在其野生栖息地的生存,因为很难同时招募颤抖和协调的战斗或逃跑反应。因此,NST需要在从颤抖状态过渡到寒冷适应状态的过程中被激活。感谢您的询问,我删除了模棱两可的短语,“颤抖一次只持续几秒钟”。我同意你的看法,NST在鸟类中的有争议的存在不能用来证明哺乳动物中肌肉NST的存在,我们在“介绍”部分添加了这句话。
- 作者继续引用curare实验,尽管我说给予的剂量显然不足以防止肌肉活动(包括颤抖)。给予足够蹒跚的动物必须通风,不能移动。与文章相关的电影表明情况并非如此,因此该实验无效。在这种情况下,它不能用作参数。这篇论文被《自然》杂志接受并不能证实不正确的实验。对本文的引用必须至少包括未使用足够剂量的curare的信息。
回复:很抱歉我没有想清楚剂量。我同意你的观点,即给予的剂量不足以阻止肌肉活动。因此,我修改了稿件,增加了“足够剂量”的内容,并将“最小化”一词改为“减少”,以使表达更加准确。
- The authors have attempted to calculate the total UCP1 in the “other” BAT depots but seem to have misunderstood the task. The important value is the total UCP1 amount in the CR animals versus the CS animals, not versus any animals in the warm. Please attempt again to recalculate. A total amount cannot be per gram, it must be per total tissue (amount per gram multiplied with tissue weight)
Reply: I'm sorry, I really misunderstood your meaning in the last modification. This time, I made a new modification. The total UCP1 content in CR group and CS group was recalculated, and the UCP1 content per gram of tissue was multiplied by tissue weight to obtain the final total UCP1 content. And the corresponding "results" section has been modified.

Reviewer 2 Report
Thank you for addressing the comments. There are still some points that haven't been fully addressed:
Comments 1 & 2: You have explained to me the advantage and implications of the study, but these have to be included in the manuscript as well (in the discussion section).
Food intake: it is not clear what is meant by "many small mammals reduce their calorie intake by losing weight". Please rephrase. Also, add a reference for this study.
"Please adjust superscripts & remove [Error! Bookmark not defined]": In your PDF version, many citation numbers have been replaced with an error. However, I am sure this would be addressed with reviewing.
Author Response
Comments 1 & 2: You have explained to me the advantage and implications of the study, but these have to be included in the manuscript as well (in the discussion section).
Reply: According to your suggestion, we have added the content about the advantages and impacts of wild rodents to the "Discussion" section. We added “Small wild rodents are natural endotherm species. When faced with natural seasonal environmental temperature fluctuations, compared with ectotherms such as nematodes, zebrafish, and model animal such as rats and mice, wild animals' response to environmental temperature changes is more similar to that of human beings. Therefore, it has more reference value for the study of human thermogenesis and thermoregulation. In addition, using wild animals instead of model animals can highlight the adaptability of animals to the environment ” and “The main purpose of this study was to determine whether the skeletal muscles of Brandt’s voles produce heat at low temperature and whether they can compensate for thermogenesis when BAT function is minimized.”
Food intake: it is not clear what is meant by "many small mammals reduce their calorie intake by losing weight". Please rephrase. Also, add a reference for this study.
Reply: I'm sorry that I didn't make my meaning clear. I have modified this sentence to “In winter, many small mammals reduce their body's demand for heat by losing weight[45]”. In addition to the original reference, this revision adds a new literature.
45.Lovegrove BG. Seasonal thermoregulatory responses in mammals. J Comp Physiol B. 2005;175(4):231-47. doi: 10.1007/s00360-005-0477-1.
"Please adjust superscripts & remove [Error! Bookmark not defined]": In your PDF version, many citation numbers have been replaced with an error. However, I am sure this would be addressed with reviewing.
Reply: Thank you for your answer to my question. I found the problem, some words became superscripts, I have corrected them.
